

# On the importance of wind predictions in wake steering optimization

Elie Kadoche [1,2], Pascal Bianchi [2], Florence Carton [1], Philippe Ciblat [2], and Damien Ernst [2,3]

[1]TotalEnergies OneTech, 2 Place Jean Millier, 92400 Courbevoie, France
[2]Télécom Paris, 19 Place Marguerite Perey, 91120 Palaiseau, France
[3]Montefiore Institute, University of Liège, Belgium

**Correspondence:** Elie Kadoche (elie.kadoche@totalenergies.com)

**Abstract.** Wake steering is a technique that optimises the energy production of a wind farm by employing yaw control to misalign upstream turbines with the incoming wind direction. This work highlights the important dependence between wind direction variations and wake steering optimization. The problem is formalized over time as the succession of independent and steady-state yaw control problems. Then, this work proposes a reformulation of each steady-state problem by augmenting the objective function by a new heuristic based on a wind prediction. The heuristic acts as a penalization for the optimization, encouraging solutions that will guarantee future energy production. Finally, a synthetic sensibility analysis of the wind direction variations and wake steering optimization is conducted. Because of the rotational constraints of the turbines, as the magnitude of the wind direction fluctuations increases, the importance of considering wind prediction in a steady-state optimization is empirically demonstrated. The heuristic proposed in this work greatly improves the performance of controllers and compared to a model predictive control (MPC) approach, it does not increase complexity.

## Nomenclature

| | | | | |
|---|---|---|---|---|
| $\alpha_t^i$ | Yaw angle | | $f_{\text{simulation}}$ | Local velocities computation function |
| $\beta_t^i$ | Absolute orientation | | $f_{\text{yaw}}$ | Yaw angle computation function |
| $\Delta t$ | Time step duration | | $H$ | Horizon length |
| $\epsilon_{K,t}$ | Wind direction noise | | $K_t$ | Wind direction |
| $\epsilon_{V,t}$ | Wind speed noise | | $K_t'$ | Observed wind direction |
| $\nu_t^i$ | Local wind speed | | $N$ | Number of wind turbines |
| $\Phi$ | Theoretical power output function | | $P_t^i$ | Turbine power output |
| $\pi$ | Control policy | | $s_t$ | Controller state |
| $f_{\text{control}}$ | Absolute orientation update function | | $u_t$ | All turbine yaw settings |
| $f_{\text{obj}}$ | Controller objective function | | $u_t^i$ | Yaw setting |
| $f_{\text{power}}$ | Power output function | | $V_t$ | Wind speed |
| | | | $V_t'$ | Observed wind speed |

## 1 Introduction

As global energy consumption increases, there is a strong willingness and necessity to decarbonize electricity production. Hence, renewable energies are becoming increasingly important (Chu and Majumdar, 2012). Wind energy, particularly, is the focus of considerable research and development, with turbines becoming larger and more numerous within wind farms. Assuring efficient control as wind turbines operate is necessary to maximize the benefits of wind energy.



In the particular context of global warming, designing more efficient wind farms is essential. Wake steering is the subject of growing interest within the community to optimize the energy production of wind farms. However, most of research regarding wind farm control technologies disregards the relevance of the wind direction. This work is motivated by a central question: from what magnitude of wind direction fluctuations is it necessary to consider the wind dynamics in steady-state wake steering optimization? To answer this question, this work proposes a new controller based on wind predictions and conducts a synthetic

sensibility analysis of wake steering and wind dynamics, using steady-state models and artificial wind data.

## 1.1 Wake effect

A single wind turbine reaches its maximum power output when fully aligned with the wind. When the wind direction changes, a turbine uses its yaw to rotate its nacelle on a horizontal plane. By using active yaw control, a wind turbine can keep track of the changes in the wind direction changes and ensure maximum energy production over time by minimizing its misalignment

with the wind. It corresponds to greedy control, where a wind turbine solely tries to maximize its power output (Yang et al., 2021).

In the space immediately behind a turbine, the wind speed is slower and more turbulent. Such a phenomenon is called the "wake effect" and is the natural consequence of wind power extraction by the machine. When a wind turbine is located in the wake of another, its power output is reduced (because of a slower wind speed) and its fatigue increased (because of the

turbulence). Within a wind farm, depending on the wind direction and the farm layout, most of the turbines can be affected by the wake of others.

Because of wake effects, greedy control can be suboptimal within a farm. Therefore, instead of keeping every turbine aligned with the wind, yaw control can also be used to voluntary misalign some turbines in relation to the direction of the wind (Boersma et al., 2017). When a turbine is misaligned with the wind, its wake effect is steered. By intelligently yawing the

turbines and steering the wake effects, the wind flow across the turbines can be optimized. Such a method is known as wind farm flow control (WFFC) (Meyers et al., 2022). A simple example on a two-turbine wind farm is given in Figure 1.

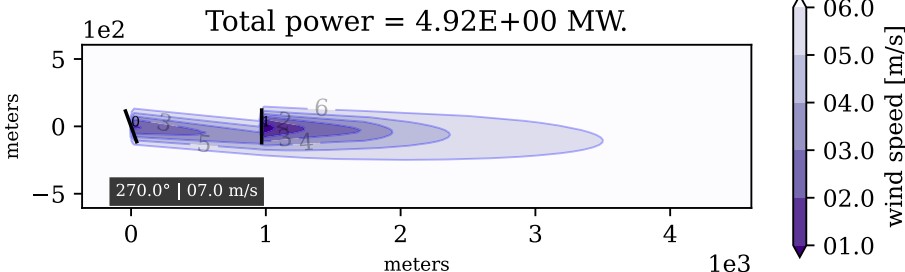

**Figure 1.** Example of WFFC on a two-turbine wind farm with the wind coming from the west. The first (upstream) turbine is misaligned and its wake effect is steered away from the second (downstream) turbine. By letting the wind flow more freely to the second turbine, the misalignment of the first turbine increases the total power output of the farm.





Current implemented wake steering strategies usually involve lookup tables (LUTs) (Fleming et al., 2017; Siemens Gamesa Renewable Energy, 2019). Wake steering strategies are computed for a finite set of different wind conditions prior to the farm operation. The yaw angles of each turbine are computed with steady-state models, regardless of the wind dynamics. Because a wake steering strategy creates misalignment with the wind, it is highly dependent on variations in the direction of the wind. The wind direction can change over time, and yaw control is constrained by the limited rotational speed of the nacelles. If the wind varies in directions and frequencies that the yaw actuators can not easily track, computing adequate wake steering strategies over time can be a challenging task.

## 1.2 Wind direction dynamics

The study of wind direction dynamics is gaining interest within the research community. Wind direction dynamics can be broken down into large-scale drifts and small-scale fluctuations (van Doorn et al., 2000) and can be observed on different scales: the synoptic scale describes long distances and extended time periods, the mesoscale depicts the farm level and time periods from days to weeks, and the microscale corresponds to the turbine level and variations from seconds to minutes. The wind direction is fundamentally non-stationary, and there is an incomplete knowledge regarding the physical and statistical characteristics of wind direction fluctuations across specific length and time scales that are essential for effective WFFC (Dallas et al., 2023).

As the farm operates, the wind direction varies both in time (at the farm level) and space (at the turbine level). The authors von Brandis et al. (2023) found that spatial wind direction changes relevant to the operation of wind farm clusters in the German Bight exceed 11 degrees in 50 % of cases. In this present work, numerical simulations are run with steady-state wake models. Therefore, only variations in the wind at the farm level are studied. When the direction varies over time, this work considers that it affects the whole wind farm.

WFFC is most beneficial at low wind speeds because this is where small changes of the wind speeds can lead to important power output variations. The same wake steering strategy will lead to higher power gains at low speeds compared to at a higher wind speed. Because the wind direction variability is higher for low wind speeds (von Brandis et al., 2023; van Doorn et al., 2000; Dallas et al., 2023), the study of dependence between wind direction variations and yaw control is important. Also, because the impact of climate change on wind dynamics is unknown, designing robust controllers is necessary for long-term operation.

## 1.3 Related works

As tracking wind direction is essential for wind turbines, the literature is rich in studies seeking better wind direction tracking mechanisms. Song et al. (2018) developed an MPC-based controller on a finite control set to track the wind directions. Hure et al. (2015) designed a yaw controller based on very short-term wind predictions. But performing WFFC and wake steering is a more complex optimization problem.

LUTs can be adapted for dynamic control with different methods. Usually, a low-pass filter is used to apply control only for high variations of the direction. A sampling method can be used to adjust the yaw control frequency and hysteresis mechanisms



avoid unnecessary yaw control and restrict the yaw actuators (Kanev, 2020a). Simley et al. (2021) improved a traditional LUT by anticipating the wind direction changes ahead of upstream turbines. Kanev (2020b) performed WFFC with receding horizon using gradient-based optimization and run tests in large-eddy simulations under realistic variations in wind direction and speed. But the wake steering strategies of an LUT fundamentally do not consider the wind dynamics, only their implementation does.

Regarding machine learning (ML) methods, and more particularly reinforcement learning (RL), which is becoming a source

of great interest to the scientific community, wind direction variations are often overlooked. The importance of the wind direction dynamics is clearly pointed out by Saenz-Aguirre et al. (2019) and Saenz-Aguirre et al. (2020) but most of the studies carried out later only consider static or quasi-static wind directions. Some recent works have started to consider dynamic wind directions into WFFC optimization (Kadoche et al., 2023).

### 1.4   Contributions

The remainder of this paper is structured to mirror the three main contributions. Each contribution forms the basis of an individual Section and Section 5 concludes. The contributions, and their corresponding Sections, are as follows.

- This work proposes a discretized formalization of the WFFC problem over time as the succession of multiple steady-state optimization problems. Due to the rotational constraints of the turbines, the important hypotheses regarding the transition between one steady-state to the next are formulated. This formalization is conducted in Section 2.

- This work presents a reformulation of the instantaneous, steady-state optimization problem. The default objective function is augmented by a new heuristic, computed on a prediction of the wind. The proposed heuristic acts as a penalization for the optimization without increasing its dimension and encourages solutions that will guarantee future energy production. The heuristic and the other studied controllers are detailed in Section 3.

- This work conducts a sensibility analysis of the wind direction variations and wake steering optimization. It empirically

demonstrates the importance of a wind prediction-based control when the magnitude of the wind direction fluctuations become large. The new proposed heuristic greatly improves the performance of a traditional steady-state wake steering optimization when the variations of the wind direction are important. Numerical simulations using synthetic wind data are conducted in Section 4.

## 2   Problem formalization

The environment is composed of a wind farm, and some exogenous variables related to wind dynamics. The wind farm consists in $N$ interconnected wind turbines. Over time, each turbine is controlled via its yaw. An episode consists of a succession of $H$ time steps during which the turbines are controlled, with $H$ the horizon length. The transition from a time step $t$ to a time step $t+1$ corresponds to a specific time window of constant length of $\Delta t$ minutes. The environment evolves from one time step $t$ to the next time step $t+1$ in $\Delta t$ minutes, with $t \in \{0, 1, \ldots, H-1\}$.



## 2.1 Wind dynamics

At a time step $t$, the exogenous variables are a global incoming wind direction $K_t \in [0, 360]$ [degrees] and a global incoming wind speed $V_t \in [\nu_{\min}, \nu_{\max}]$ [m/s]. The wind data can be measured or predicted. Then, a controller does not have access to $K_t$ and $V_t$ directly but to $K'_t$ [degrees] and $V'_t$ [m/s], with $K'_t$ a noisy wind direction defined as $K'_t = K_t + \epsilon_{K,t}$ and $V'_t$ is a noisy wind speed defined as $V'_t = V_t + \epsilon_{V,t}$. The random noises $\epsilon_{K,t}$ and $\epsilon_{V,t}$ can come from either measurement imprecisions or prediction errors.

Because the turbines alter the wind flow inside the farm, at a time step $t$, the wind speed in front of a turbine $i$ can be different from $V_t$ and is noted as $\nu^i_t$ [m/s]. The computation of the local velocities is based on complex fluid mechanics and is subject to numerous uncertainties. This work conducts numerical experiments where such computations are done using a low-fidelity, steady-state simulator FLOw Redirection and Induction in Steady State (FLORIS) NREL (2021). The local wind directions stay equal to $K_t$, as it is common for low-fidelity, steady-state simulation.

## 2.2 Turbines

At a time step $t$, a turbine $i$ is characterized by its absolute angular position $\beta^i_t \in [0, 360]$ [degrees] and its relative orientation or yaw (often used to compute the power output) $\alpha^i_t = f_{\text{yaw}}(K_t, \beta^i_t) \in [-180, 180]$ [degrees], such that

$$f_{\text{yaw}}(K_t, \beta^i_t) = (K_t - \beta^i_t + 180) \mod 360 - 180. \tag{1}$$

Adding and subtracting by 180 ensures that the yaw stays in the range $[-180, 180]$. As illustrated in Figure 2, the yaw corresponds to the rotational movement going from the absolute angular position $\beta^i_t$ to the wind direction $K_t$, such that $\beta^i_t + \alpha^i_t \mod 360 = K_t$. Positive values of the yaw indicate that the turbine is rotated anti-clockwise from the wind direction and negative values of the yaw indicate that the turbine is rotated clockwise from the wind direction.





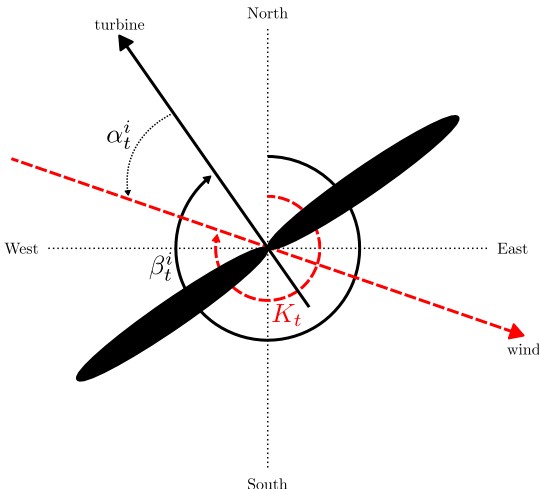

**Figure 2.** Example of a wind turbine $i$ seen from above at a time step $t$. The variables are the wind direction $K_t$, the absolute angular position $\beta_t^i$ of the turbine, and the yaw $\alpha_t^i$ of the turbine. The wind direction indicates from where the wind is coming, e.g., a wind direction of 270 degrees indicates a wind coming from the west. Here, the nacelle is misaligned with the incoming wind direction: the turbine is rotated clockwise from the wind, so $\alpha_t^i < 0$.

At a time step $t$, the yaw setting $u_t^i \in [u_{\min}, u_{\max}]$ [degrees] of a turbine $i$ corresponds to the rotational movement of the turbine between time steps $t$ and $t+1$. Because the nacelle has rotational constraints, the yaw setting is bounded between two consecutive time steps. The setting is used to update the orientation of the turbine $\beta_{t+1}^i = f_{\text{control}}(\beta_t^i, u_t^i)$ such that

$$f_{\text{control}}(\beta_t^i, u_t^i) = (\beta_t^i + u_t^i) \mod 360. \tag{2}$$

### 2.3 Power

For any wind speed $\nu$ [m/s], the theoretical power output [megawatts (MW)] of a turbine considering no yaw misalignment is given by the power curve $\Phi$, such that

$$\Phi(\nu) = \frac{1}{2} \cdot \rho \cdot A \cdot \nu^3 \cdot C_P(\nu), \tag{3}$$

with $\rho$ [kg/m$^3$] the air density, $A$ [m$^2$] the rotor blade area and $C_P$ the power coefficient of the turbine. The theoretical power output is strictly positive if the wind speed is within certain bounds $[\nu_{\text{cut-in}}, \nu_{\text{cut-out}}]$ [m/s]. An illustration is given in Figure 3.

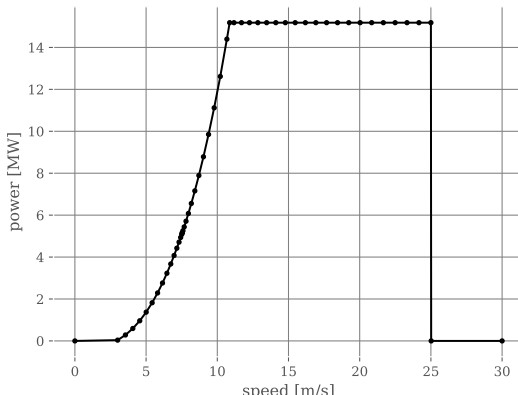

**Figure 3.** Example of the International Energy Agency (IEA) 15 MW wind turbine power curve (Gaertner et al., 2020). The power curve $\Phi$ gives the theoretical power output (y-axis) of the machine as a function of the wind speed (x-axis), considering no yaw misalignment. Here, $\nu_{\text{cut-in}} = 3$ m/s and $\nu_{\text{cut-out}} = 25$ m/s.

At a time step $t$, the power output of a turbine $i$ considering yaw misalignment $P_t^i = f_{\text{power}}(\nu_t^i, \alpha_t^i)$ [MW] is computed from the power curve and the yaw angle, such that

$$f_{\text{power}}(\nu_t^i, \alpha_t^i) = \Phi(\nu_t^i) \cdot \cos^p(\alpha_t^i) \cdot \mathbb{1}_{\{\alpha_{\text{cut-in}} \leqslant \alpha_t^i \leqslant \alpha_{\text{cut-out}}\}}, \tag{4}$$

with $p$ a parameter accounting for power losses due to misalignment and $[\alpha_{\text{cut-in}}, \alpha_{\text{cut-out}}]$ [degrees] a safety bound for the yaw. Because too much misalignment with the wind can damage the machine, if the yaw is too great, the turbine is shut down and its power output is null.

### 2.4 Policy

A policy $\pi$ is a function returning the yaw settings $(u_t^0, u_t^1, \ldots, u_t^{N-1})$ of all the turbines at a time step $t$ given a state $s_t$. Each wind farm controller is associated with a specific policy. In this work, the state $s_t$ may be composed of an observation of the current wind, a prediction of the wind on $L$ time steps and the current orientations of the turbines. The general form of a state is

$$s_t = \{K_t', V_t', K_{t+1}', V_{t+1}', \ldots, K_{t+L}', V_{t+L}', \{\beta_t^i\}_{i \in \{0,1,\ldots,N-1\}}\}. \tag{5}$$

States can be categorized based on two distinct properties: with perfect or imperfect information and with or without foresight knowledge. Depending on the possible combinations, there are four classes of states, listed in Table 1.



| | foresight | no foresight |
|---|---|---|
| perfect information | $\epsilon_{K,t+k} = 0$ and $\epsilon_{V,t+k} = 0, L > 0$ | $\epsilon_{K,t+k} = 0$ and $\epsilon_{V,t+k} = 0, L = 0$ |
| imperfect information | $\epsilon_{K,t+k} \neq 0$ or $\epsilon_{V,t+k} \neq 0, L > 0$ | $\epsilon_{K,t+k} \neq 0$ or $\epsilon_{V,t+k} \neq 0, L = 0$ |

**Table 1.** Four different classes of states based on two distinct properties, with $t \in \{0, 1, \ldots, H-1\}$ and $k \in \{0, 1, \ldots, L\}$. With perfect information, the state comprises exact wind data (no noise). Without foresight, the state only comprises the current wind data (no predictions).

## 2.5 System dynamics

An episode is defined by $H$ time steps during which turbines are controlled via their yaw. An episode is characterized by time series for the wind directions and the wind speeds and initial positions for the nacelles. During an episode, it is assumed that all the states belong to the same class (defined in Table 1) and the policy is presumed to be stationary (it does not change over time).

The full dynamics of an episode are described in Algorithm 1. At each time step, the policy returns the yaw settings based 175 on the current state, the system is updated and the power output of the farm is computed. The yaw setting of a turbine $i$ at the end of time step $t$ is indexed $t+1$ (because it has been updated) and it is the one used for the power computation of time step $t$.

At a time step $t$, to compute the power output of each turbine, local wind velocities are needed. Such computations rely on complex fluid mechanics, depending on the incoming wind and the updated yaw angles of each turbine. In this work, these computations are carried out by a steady-state simulator $\nu_t^i = f_{\text{simulation}}^i(K_t, V_t, \{\alpha_{t+1}^j\}_{j \in \{0,1,\ldots,N-1\}})$, which is used as 180 a substitute for real-life measurements. The simulation is said to be steady-state because it only depends on the current wind data and the updated yaw angles. It does not consider time delays in the wake propagation.

---

**Algorithm 1** Full episode dynamics over time

**Input:** $\{K_k, V_k\}_{k \in \{0,1,\ldots,H+L-1\}}$ true wind data time series
$\qquad\{\beta_0^i\}_{i \in \{0,1,\ldots,N-1\}}$ initial orientations
$\qquad\pi$ yaw control policy
**for** $t = \{0, 1, \ldots, H-1\}$ **do**
$\qquad s_t = \{K_t', V_t', K_{t+1}', V_{t+1}', \ldots, K_{t+L}', V_{t+L}', \{\beta_t^i\}_{i \in \{0,1,\ldots,N-1\}}\}$ $\qquad\qquad\qquad$ ▷ state
$\qquad (u_t^0, u_t^1, \ldots, u_t^{N-1}) = \pi(s_t)$ $\qquad\qquad\qquad\qquad\qquad\qquad\qquad\qquad$ ▷ control policy
$\qquad \beta_{t+1}^i = f_{\text{control}}(\beta_t^i, u_t^i), \forall i \in \{0, \ldots, N-1\}$ $\qquad\qquad\qquad\qquad$ ▷ angular positions update
$\qquad \alpha_{t+1}^i = f_{\text{yaw}}(K_t, \beta_{t+1}^i), \forall i \in \{0, \ldots, N-1\}$ $\qquad\qquad\qquad\qquad$ ▷ yaw angles computation
$\qquad \nu_t^i = f_{\text{simulation}}^i(K_t, V_t, \{\alpha_{t+1}^j\}_{j \in \{0,1,\ldots,N-1\}}), \forall i \in \{0, \ldots, N-1\}$ $\qquad$ ▷ local velocities computation
$\qquad P_t^i = f_{\text{power}}(\nu_t^i, \alpha_{t+1}^i), \forall i \in \{0, \ldots, N-1\}$ $\qquad\qquad\qquad\qquad$ ▷ power outputs computation
**end for**
**Output:** $\sum_{t=0}^{H-1} \sum_{i=0}^{N-1} P_t^i$ farm power output

---





At each time step, during the "control policy" operation, a controller can conduct any computations with the $f_\text{control}, f_\text{yaw}, f_\text{simulation}$ and $f_\text{power}$ functions but based on the wind data provided by the state. Because such data can be noisy, all the computed values can be inexact. For example, at a time step $t$, if a controller computes the yaw of a turbine $i$ based on its updated orientation $\beta_{t+1}^i$, it would be equal to $\alpha_{t+1}^i{}' = f_\text{yaw}(K_t', \beta_{t+1}^i)$. Because the observed wind direction $K_t'$ can be different from the true wind direction $K_t$, the estimated yaw $\alpha_{t+1}^i{}'$ can be different from its true value $\alpha_{t+1}^i$.

## 2.6 Transition regime

At a time step $t$, for a turbine $i$, the WFFC problem thus formalized considers a single power output $P_t^i$. In reality, during a time step, the wind is time varying and a turbine takes time to rotate because of mechanical constraints. Therefore, the discretization of the continuous control problem results in the loss of some information and possibly less inaccurate power outputs. To ensure that the discretized power outputs are good approximations, from one time step to another, a turbine is supposed to rotate immediately and the wind is supposed to be quasi-constant.

The duration of a time step is always considered constant during an episode. At a time step $t$, when a setting $u_t^i$ is applied to a turbine $i$, the rotational time $T_r$ [minutes] for the turbine to go from its current orientation $\beta_t^i$ to its next orientation $\beta_{t+1}^i$ is always considered largely inferior to the duration of the time step, i.e., $T_r \ll \Delta t$ for all $u_t^i \in [u_\text{min}, u_\text{max}]$. A turbine always reaches rapidly its target position, before the end of the time step duration. But during a time step, no other control will be applied to the turbine. For this reason, the rotational constraints $[u_\text{min}, u_\text{max}]$ need to be consistent with the duration of a time step $\Delta t$.

The coherence time $T_c$ [minutes] of a wind variable (either the direction or the speed) is the maximum duration during which the variable is quasi-constant. If the coherence time of the wind direction is strictly smaller than the time step duration, a discretized value $K_t$ would stretch too far away from its corresponding continuous signal. The same goes for the speed. Therefore in this work, the coherence time is always equal to the time step duration, i.e., $T_c = \Delta t$, for both the direction and the speed.

## 3 Controllers

At a time step $t$, the yaw settings $(u_t^0, u_t^1, \ldots, u_t^{N-1})$ are noted $u_t$. A controller is defined by its policy $\pi(s_t)$ with the state $s_t$ given by Equation (5). This work compares a naive control where each turbine is aligned as much as possible with the wind and three optimized wake steering control strategies. In a episode, at each time step $t$, during the "control policy" operation of Algorithm 1, each controller computes the yaw settings such that $u_t = \pi(s_t)$, by maximizing a specific objective function $f_\text{obj}(s_t, u_t)$ with regard to the turbine rotational constraints, as defined below.

$$\pi(s_t) \in \underset{u_t}{\arg\max}\, f_\text{obj}(s_t, u_t), \tag{6}$$

$$\text{subject to} \quad u_\text{min} \leqslant u_t^i \leqslant u_\text{max}, \; \forall\, t \in \{0, 1, \ldots, H-1\}, \; \forall\, i \in \{0, 1, \ldots, N-1\}. \tag{7}$$





## 3.1 Naive controller

The naive controller always tries to keep turbines aligned with the current wind direction as much as possible. It is a weak
baseline as it does not conduct any wake steering optimization. It runs with no foresight (i.e., $L = 0$), as it is only concerned
with the current observed wind direction $K'_t$. Therefore, the state $s_t$ is reduced to $\{K'_t, \{\beta^i_t\}_{i \in \{0,1,\dots,N-1\}}\}$. It consists of a
greedy control (no wake steering) where the objective function at a time step $t$ is minimizing the amplitude of the yaws, i.e.,

$$f_{\text{obj}}(s_t, u_t) = -\sum_{i=0}^{N-1} |\alpha^i_{t+1}{}'|, \tag{8}$$

$$\text{with} \quad \alpha^i_{t+1}{}' = f_{\text{yaw}}(K'_t, \beta^i_{t+1}), \tag{9}$$

$$\beta^i_{t+1} = f_{\text{control}}(\beta^i_t, u^i_t). \tag{10}$$

At a time step $t$, the rotational movement required for a turbine $i$ to stay aligned with the observed wind direction is $f_{\text{yaw}}(K'_t, \beta^i_t)$.
Because of the rotational constraints, this movement is clipped, giving a closed-form expression for the solution, defined as

$$\pi_{\text{naive}} = (u^0_t, u^1_t, \dots, u^{N-1}_t), \tag{11}$$

$$\text{such that} \quad u^i_t = \text{clip}(f_{\text{yaw}}(K'_t, \beta^i_t), u_{\min}, u_{\max}), \, \forall \, i \in \{0, 1, \dots, N-1\}. \tag{12}$$

## 3.2 Wake steering

Compared to naive control, wake steering is used to optimize the power output of the farm. In this work, two distinct steering
strategies are used. First, an instantaneous optimization searches for the yaw settings maximizing the instantaneous power
output of the farm. Secondly, an MPC like approach considers prediction of the wind in its optimization. At each time step $t$,
the same Gauss-Seidel (GS) method is used. The objective function varies across the two different controllers. In this work,
optimization is conducted with a GS method (described in Algorithm A1 of Appendix A). A similar approach was first proposed
by Fleming et al. (2022) with a serial-refine algorithm.

A first solution is initialized from the naive controller, each initial yaw setting keeping its turbine aligned as much as possible
with the wind. Then, the GS method iterates over each turbine, from upstream to downstream ones. At each iteration, it solves
the optimization problem for the current turbine, considering the yaw settings of all others fixed. To do so, it uses a grid-search
method over a discretized solution space $S = \{u_{\min} + l \cdot \frac{u_{\max} - u_{\min}}{n_y - 1}\}$, for all $l \in \{0, 1, \dots, n_y - 1\}$ with $n_y$ being a precision
parameter. Once optimized, the setting of the current turbine is updated, and it moves to the next one.

### 3.2.1 Instantaneous controller

The instantaneous controller searches for the yaw settings maximizing the immediate power output of the farm. It always runs
under no foresight (i.e., $L = 0$), as it performs wake steering for the current, observed wind data only. Therefore, the state $s_t$
is reduced to $\{K'_t, V'_t, \{\beta^i_t\}_{i \in \{0,1,\dots,N-1\}}\}$. It is a steady-state optimization performed on one time step where the objective



function at a time step $t$ is the immediate normalized power output, i.e.,

$$f_{\text{obj}}(s_t, u_t) = \frac{1}{N} \cdot \sum_{i=0}^{N-1} f_{\text{power}}(\nu_t^{i'}, \alpha_{t+1}^{i}{}'), \tag{13}$$

$$\text{with} \quad \nu_t^{i'} = f_{\text{simulation}}^{i}(K_t', V_t', \{\alpha_{t+1}^{j}{}'\}_{j \in \{0,1,\ldots,N-1\}}), \tag{14}$$

$$\alpha_{t+1}^{i}{}' = f_{\text{yaw}}(K_t', \beta_{t+1}^{i}), \tag{15}$$

$$\beta_{t+1}^{i} = f_{\text{control}}(\beta_t^{i}, u_t^{i}). \tag{16}$$

### 3.2.2 Prediction-based controller

A traditional prediction-based controller searches for the yaw settings of time steps $t, t+1, \ldots, t+L$ that maximize the power output over that horizon. It always runs with foresight (i.e., $L \geqslant 1$). To optimize the yaw settings over a time horizon, an MPC method is commonly used. The corresponding optimization problem is the following.

$$\max_{u_t, u_{t+1}, \ldots, u_{t+L}} \quad \frac{1}{N} \cdot \sum_{k=0}^{L} \sum_{i=0}^{N-1} f_{\text{power}}(\nu_{t+k}^{i}{}', \alpha_{t+k+1}^{i}{}'), \tag{17}$$

$$\text{subject to} \quad u_{\min} \leqslant u_{t+k}^{i} \leqslant u_{\max}, \; \forall \, t \in \{0,1,\ldots,H-1\}, \; \forall \, k \in \{0,1,\ldots,L\}, \; \forall \, i \in \{0,1,\ldots,N-1\}, \tag{18}$$

$$\text{with} \quad \nu_{t+k}^{i}{}' = f_{\text{simulation}}^{i}(K_{t+k}', V_{t+k}', \{\alpha_{t+k+1}^{j}{}'\}_{j \in \{0,1,\ldots,N-1\}}), \tag{19}$$

$$\alpha_{t+k+1}^{i}{}' = f_{\text{yaw}}(K_{t+k}', \beta_{t+k+1}^{i}), \tag{20}$$

$$\beta_{t+k+1}^{i} = f_{\text{control}}(\beta_{t+k}^{i}, u_{t+k}^{i}). \tag{21}$$

The MPC thus described multiplies the number of decision variables by $L$. Also, the computation of the local velocities at a given time step depends on all the previous yaw settings. Therefore, the MPC significantly increases the complexity, and because there is no simple solution, this work proposes a reformulation. The objective function given by Equation (17) can be split between the current time step $t$ and the next ones, from $t+1$ to $t+L$, such that

$$\frac{1}{N} \cdot \sum_{k=0}^{L} \sum_{i=0}^{N-1} f_{\text{power}}(\nu_{t+k}^{i}{}', \alpha_{t+k+1}^{i}{}') = \frac{1}{N} \cdot f_{\text{power}}(\nu_t^{i}{}', \alpha_{t+1}^{i}{}') + \frac{1}{N} \cdot \sum_{k=1}^{L} \sum_{i=0}^{N-1} f_{\text{power}}(\nu_{t+k}^{i}{}', \alpha_{t+k+1}^{i}{}'). \tag{22}$$

The first term of Equation (22) is the normalized power output of the farm for the current time step. It corresponds to the objective function of the instantaneous controller defined in Subsubsection 3.2.1. It only depends on the current yaw settings $u_t$. Now, focusing on the second term, the closed-form expression of the $f_{\text{power}}$ is written, giving

$$\frac{1}{N} \cdot \sum_{k=1}^{L} \sum_{i=0}^{N-1} f_{\text{power}}(\nu_{t+k}^{i}{}', \alpha_{t+k+1}^{i}{}') = \frac{1}{N} \cdot \sum_{k=1}^{L} \sum_{i=0}^{N-1} \Phi(\nu_{t+k}^{i}{}') \cdot \cos^{p}(\alpha_{t+k+1}^{i}{}') \cdot \mathbb{1}_{\{\alpha_{\text{cut-in}} \leqslant \alpha_{t+k+1}^{i}{}' \leqslant \alpha_{\text{cut-out}}\}}. \tag{23}$$

The complexity brought by the MPC comes from the fact that Equation (23) depends on the local velocities $\nu_{t+k}^{i}{}'$ and the updated yaw angles $\alpha_{t+k+1}^{i}{}'$ corresponding to the optimized yaw settings of each future time step. To decrease the complexity, this work proposes to modify Equation (23) in the following way.





– Each local velocity $\nu_{t+k}^{i}{}'$ is replaced by the corresponding, predicted, global wind speed $V_{t+k}'$. It reduces the complexity coming from the steady-state simulation by removing the dependence with the updated yaw angles.

– Each updated yaw angle $\alpha_{t+k+1}^{i}{}'$ depending on the optimized yaw setting $u_{t+k}^{i}$ is replaced by the expected yaw angle $\hat{\alpha}_{t+k+1}^{i}{}'$ if a naive controller was used instead. It reduces the complexity coming from the optimization, as there is a closed-from expression for the naive controller, as provided by Equation (11).

– The cosine function at power $p$ of each yaw angle is replaced with a simpler penalization for yaw misalignment. The penalization chosen corresponds to one minus the normalized, absolute value of that yaw angle. It provides linearity and better interpretability.

– The indicator function is removed so that there is no discontinuity. Even if a yaw is too great, it can be of some interest for the optimization to know about the potential power output. The more a turbine is misaligned, the less likely it will be to produce energy and the more it will be penalized.

– For each time step, the overall expression is multiplied by a discounted factor $\gamma \in [0, 1]$. It gives more importance to values closed to the current time step.

The only variables specific to each turbine are the yaw angles updated from a naive controller, which are already normalized. Therefore, it becomes unnecessary to normalize the overall expression by $N$. With such modifications, Equation (23) becomes a new heuristic $\mathcal{H}_t$ defined as

$$\mathcal{H}_t(s_t, u_t) = \sum_{k=1}^{L} \gamma^{k-1} \cdot \Phi(V_{t+k}') \cdot \left(1 - \frac{1}{N} \cdot \frac{1}{180} \cdot \sum_{i=0}^{N-1} |\hat{\alpha}_{t+k+1}^{i}{}'|\right), \tag{24}$$

with   $\hat{\alpha}_{t+k+1}^{i}{}' = f_{\text{yaw}}(K_{t+k}', \hat{\beta}_{t+k+1}^{i}),$     (25)

$\hat{\beta}_{t+k+1}^{i} = f_{\text{control}}(\hat{\beta}_{t+k}^{i}, \hat{u}_{t+k}^{i}),$   $\hat{u}_{t+k}^{i}$ computed with a naive controller defined by Equation (11),     (26)

    $\hat{\beta}_{t+1}^{i} = f_{\text{control}}(\beta_t^i, u_t^i),$   $u_t^i$ computed from a wake steering optimization.     (27)

Because this new proposed heuristic does depend on neither the future optimized yaw settings (naive control), nor the future local velocities (no simulation), it does not increase complexity. The heuristic is a scalar acting as a penalization for the optimization. The final objective function of the prediction-based controller can finally be written as

$$f_{\text{obj}}(s_t, u_t) = \frac{1}{N} \cdot \sum_{i=0}^{N-1} f_{\text{power}}(\nu_t^{i}{}', \alpha_{t+1}^{i}{}') + \mathcal{H}_t(s_t, u_t), \tag{28}$$

with   $\nu_t^{i}{}' = f_{\text{simulation}}^{i}(K_t', V_t', \{\alpha_{t+1}^{j}{}'\}_{j \in \{0,1,\ldots,N-1\}}),$     (29)

    $\alpha_{t+1}^{i}{}' = f_{\text{yaw}}(K_t', \beta_{t+1}^{i}),$     (30)

    $\beta_{t+1}^{i} = f_{\text{control}}(\beta_t^i, u_t^i),$     (31)

    $\mathcal{H}_t(s_t, u_t)$ defined by Equation (24).     (32)





The heuristic is the discounted, weighted sum of the future theoretical power outputs. By choosing certain optimized yaw settings $u_t$ for the current time step, the heuristic uses a naive controller over a future time horizon of $L$ time steps to evaluate how well the turbines will manage to stay aligned with the predicted wind directions.

For example, if the future expected power outputs are high, the heuristic will encourage yaw settings that will put the turbines in good orientations for the future. The heuristic will penalize the objective function for yaw settings that will prevent turbines 300  from keeping track of the wind. An illustration of the heuristic is given in Figure 4.

### 3.3 Upper bound

To have an upper bound in terms of performance (power output) of a wake steering strategy, the rotational constraints are relaxed. It means that in Equation (7), the variables $u_{\min}$ and $u_{\max}$ are equal to -180 and 180 degrees, respectively. Between two consecutive time steps, each turbine is assumed to be capable of reaching any orientation.

The same objective function of the instantaneous controller, presented in Subsubsection 3.2.1 is used. It always runs under no foresight (i.e., $L = 0$), as it performs wake steering for the current wind data only. Therefore, the state $s_t$ is reduced to $\{K'_t, V'_t, \{\beta^i_t\}_{i \in \{0,1,...,N-1\}}\}$. The yaw settings computed by the upper bound would not be admissible in reality if the corresponding targeted orientations are too far away from the current ones.

## 4 Simulations

In Subsection 4.1 the process used to generate wind data is described and in Subsection 4.2 the experiment setting is given. Finally, the results and the empirical conclusions that can be drawn are explained in Subsection 4.3.

### 4.1 Wind data scenario

The wind data time series are artificially generated with custom Wiener processes. The wind directions $\{K_t\}_{t \in \{0,1,...,H+L-1\}}$ are computed with Algorithm 2. The wind speeds $\{V_t\}_{t \in \{0,1,...,H+L-1\}}$ are computed with Algorithm 3. To generate the time 315  series, an initial value is cumulatively incremented at each time step by variable $m_t$. Each increment $m_t$ is independently sampled from a Normal distribution of mean 0 and standard deviation $\sigma_t$. The standard deviation $\sigma_t$ at each time step is equal to $\tau \times \sqrt{\delta^X_t}$ with $\tau$ a normalization variable with regard to the number and range of the generated values and $\delta^X_t$ a variation parameter for the wind variable $X$ (either the direction or the speed).





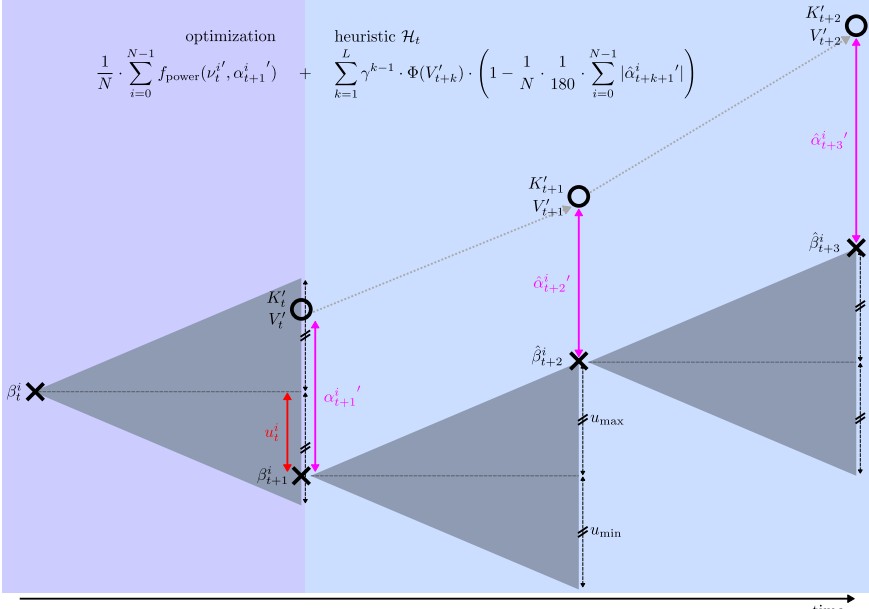

(a) In this case, the yaw setting computed at the current time step puts the turbine in a bad orientation for the future. By using a naive controller, it is still far away from future wind directions.

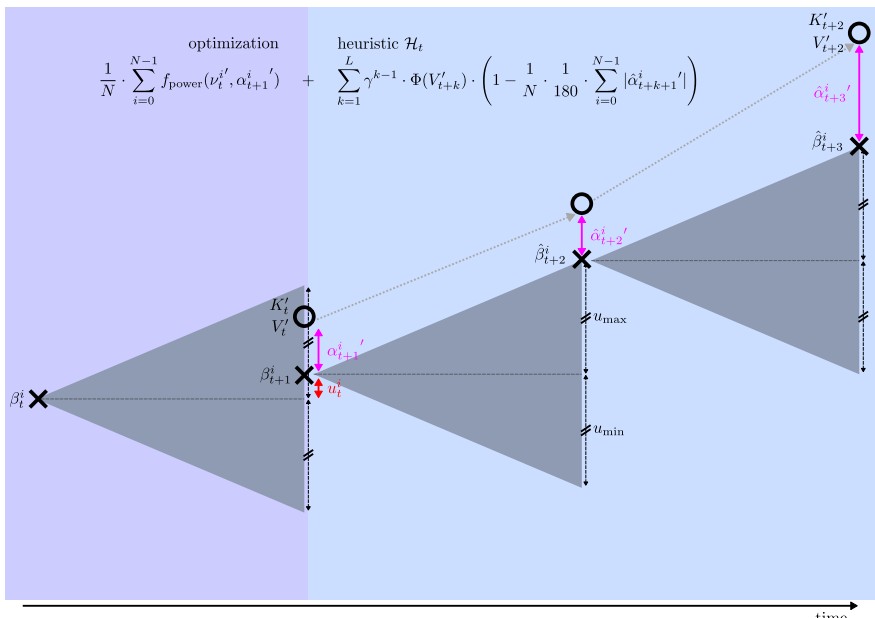

(b) In this case, the yaw setting computed at the current time step puts the turbine in a better orientation for the future. By using a naive controller, it manages to stay closer to future wind directions.

**Figure 4.** Illustration of the objective function with the new heuristic for a turbine $i$ at a time step $t$ with $L = 3$. It comprises two steps: the wake steering optimization and the heuristic. Optimization is performed to find the setting $u_t^i$ leading to the orientation $\beta_{t+1}^i$. Then, the heuristic uses a naive controller to compute the next expected yaw angles. Each triangle corresponds to the range of admissible yaw settings. The heuristic encourages the choice of yaw settings that may not be the best at the current time step but that ensure future power output.





---

**Algorithm 2** Wind directions generator

**Input:** $H + L$ number of points

$K_{\text{init}}$ initial wind direction

$\delta_{\min}^K, \delta_{\max}^K$ bounds for the variation variable

$\tau = 360/(H + L)$

**for** $t = \{0, 1, \ldots, H + L - 1\}$ **do**

$\quad \delta_t^K \sim U(\delta_{\min}^K, \delta_{\max}^K)$

$\quad \sigma_t = \tau \times \sqrt{\delta_t^K}$

$\quad m_t \sim N(0, \sigma_t)$

$\quad K_t = (K_{\text{init}} + \sum_{i=0}^t m_i) \mod 360$

**end for**

**Output:** $\{K_t\}_{t \in \{0,1,\ldots,H+L-1\}}$ wind directions time series

---

---

**Algorithm 3** Wind speeds generator

**Input:** $H + L$ number of points

$V_{\text{init}}$ initial wind speed

$\nu_{\min}, \nu_{\max}$ bounds for the wind speed

$\delta_{\min}^V, \delta_{\max}^V$ bounds for the variation variable

$\tau = (\nu_{\max} - \nu_{\min})/(H + L)$

**for** $t = \{0, 1, \ldots, H + L - 1\}$ **do**

$\quad \delta_t^V \sim U(\delta_{\min}^V, \delta_{\max}^V)$

$\quad \sigma_t = \tau \times \sqrt{\delta_t^V}$

$\quad m_t \sim N(0, \sigma_t)$

$\quad V_t = \text{mirrored}(N_{\text{init}} + \sum_{i=0}^t m_i)$

**end for**

**Output:** $\{V_t\}_{t \in \{0,1,\ldots,H+L-1\}}$ wind speeds time series

---

To maintain the wind directions in the range of valid values, i.e., $[0, 360]$ [degrees], the modulo operation is sufficient. To maintain the wind speeds in the range of valid values, i.e., $[\nu_{\min}, \nu_{\max}]$ [m/s], a mirrored function as explained in Figure 5 is proposed. The generated values inside the wind speed bounds are not modified. The generated values outside the bounds are recursively mirrored inside the bounds.





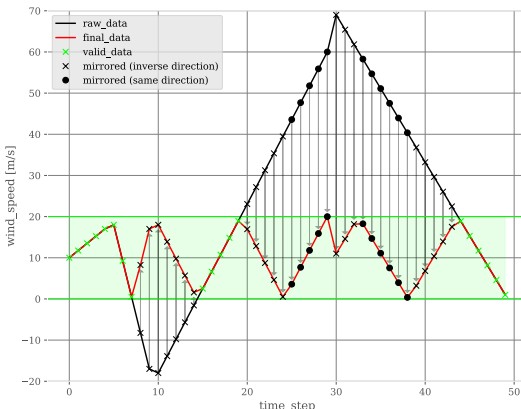

**Figure 5.** Toy example of the mirrored function used to keep the generated wind speeds inside specific bounds. Raw data is generated thanks to a process described by Algorithm 3. Raw data points inside the wind speed bounds are not modified: the black and red curves overlap. Data points outside the wind speed bounds are recursively mirrored inside the bounds.

The variable $\delta_t^X$ defines the level of variation of the wind variable $X$ time series (either the direction or the speed). When equal to 0, the signal is constant. As $\delta_t^X$ increases, the absolute value of the increments increases in average. At each time
step, $\delta_t^X$ is sampled from a uniform distribution defined between $\delta_{\min}^X$ and $\delta_{\max}^X$. When $\delta_{\min}^X$ and $\delta_{\max}^X$ are equal, all increments $\{m_t\}_{t \in \{0,1,\ldots,H+L-1\}}$ are independently sampled from the same distribution: the generated time series is stationary with regard to the increments. When $\delta_{\min}^X < \delta_{\max}^X$, increments are independently sampled from different distributions: the generated time series is non-stationary with regard to the increments. In Figure 6 the impact of the $\delta_t^K$ variable is shown for the wind direction.

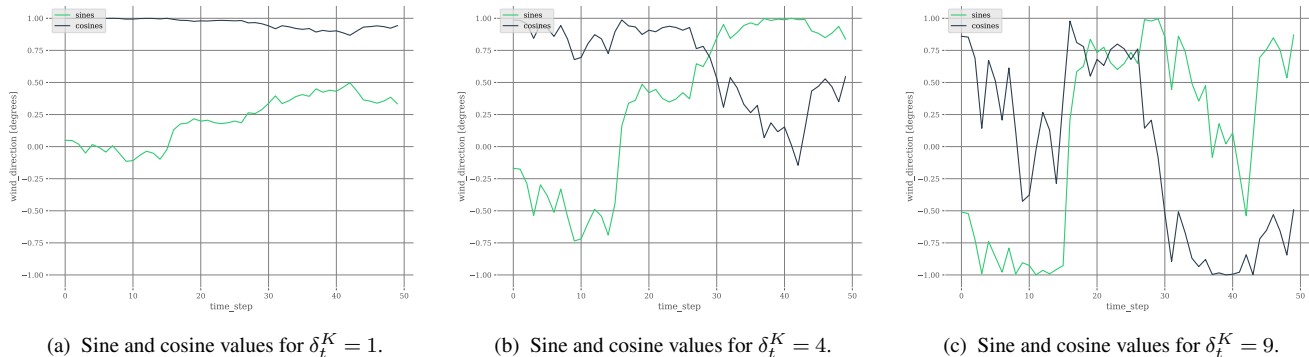

(a) Sine and cosine values for $\delta_t^K = 1$.     (b) Sine and cosine values for $\delta_t^K = 4$.     (c) Sine and cosine values for $\delta_t^K = 9$.

**Figure 6.** Example of different wind direction signals generated with different $\delta_t^K$ values, considering that $\delta_t^K = \delta_{\min}^K = \delta_{\max}^K$ for all $t \in \{0, 1, \ldots, 49\}$ and $K_{\text{init}} = 7$ degrees. If $\delta_t^K = 0$, all the generated points are equal to $K_{\text{init}}$. The sine and cosine values are plotted for illustration convenience (it avoids the discontinuity issue of degrees). Note that the behavior shown in this example is the same for the wind speed, but values are in the range $[\nu_{\min}, \nu_{\max}]$.



## 4.2 Experimental setting

The function $f^i_{\text{simulation}}(K_t, V_t, \{\alpha^j_{t+1}\}_{j \in \{0,1,\dots,N-1\}})$ computes the local wind speed in front of a turbine $i$ at a time step $t$ given wind data $K_t, V_t$ and the yaw of each turbine $\{\alpha^j_{t+1}\}_{j \in \{0,1,\dots,N-1\}}$. This function, introduced in Subsection 2.5, is assured by the low-fidelity, steady-state simulator FLORIS (NREL, 2021). FLORIS is used with a Gauss Curl hybrid wake model. The Gaussian velocity model is implemented based on Bastankhah and Porté-Agel (2016) and Niayifar and Porté-Agel (2016). To compute the deflection of the wakes depending on the yaws, the models described by Bastankhah and Porté-Agel (2016) and King et al. (2021) are used. The turbulence model described by Crespo and Herna´ndez (1996) is used. The options "secondary steering", "yaw added recovery" and "transverse velocities" are all enabled.

A wind farm of 34 IEA identical 15 MW wind turbines is used. It has cut-in and cut-out speeds of $\nu_{\text{cut-in}} = 3$ m/s and $\nu_{\text{cut-out}} = 25$ m/s, respectively. Each wind turbine has a rotor diameter of 242.24 m, i.e., a rotor area of 46087 m². The air density is $\rho = 1.225$ kg/m³ and the tunable parameter accounting for the power losses due to misalignment is $p = 1.88$. WFFC strategies are sensible to the distances between turbines. To make the numerical simulations more robust to the distances between turbines, a diamond shape is used for the layout. With a diamond shape, there is an identical distance between each machine and its surrounding turbines. 34 machines create a sufficiently large wind farm for wake steering to be impactful, and is sufficiently small for optimization to converge quickly. A FLORIS illustration of the layout used is given in Figure 7.

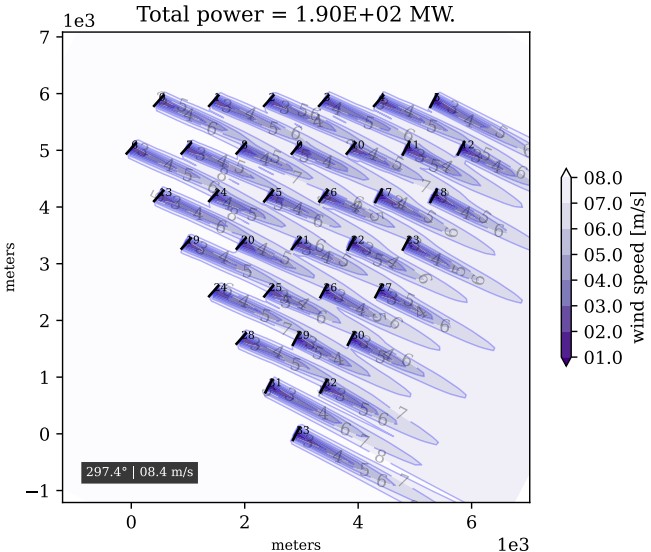

**Figure 7.** Layout in the form of a diamond shape. The farm comprises 34 identical IEA 15 MW wind turbines. There is an identical space equivalent to the diameter of four turbines between a machine and its adjacent turbines. A distance of four turbine diameters is sufficiently small to create detrimental wake effects for the farm and therefore, the optimization pertinent; and sufficient large for the design to be realistic. Here the direction is 287.4 degrees, the wind speed is 8.4 m/s and yaws are computed with the instantaneous wake steering controller.





The limits for the wind speed are $\nu_{\min} = 4$ m/s and $\nu_{\max} = 10$ m/s. The interval $[4, 10]$ m/s corresponds, approximately, to the ascending part of the power curve (Figure 3), where wake steering is the most beneficial for the farm. For wind speeds of between $[10, 25]$ m/s, the power output is constant; if the wind speed is reduced down because of wake effects, there will be no power deficit. Because this work conducts a sensibility analysis of yaw control, the wind speed is kept in the range of $[4, 10]$ m/s.

The horizon size is $H = 144$ and the length of the foresight for the prediction-based controller is $L = 11$. The initial wind values are $K_{\text{init}} = 270$ degrees and $V_{\text{init}} = 8$ m/s. The discount factor used for the heuristic $\mathcal{H}_t$ is $\gamma = 0.99$. The precision parameter for the GS methods is $n_y = 120$, giving the grid-search method good precision.

     More technical details regarding the simulations and numerical instabilities are given in Appendix B. The time step duration $\Delta t$ is intentionally undefined, as it will be explained in Subsubsection 4.3.1. Depending on the time step duration value,
different interpretations of the same results will be made.

### 4.3    Results

To empirically demonstrate the importance of optimizing yaw control over a long-term time horizon, numerical simulations are performed with perfect and imperfect (noisy) wind predictions. For each curve, the centerline corresponds to the mean and the colored area corresponds to the standard deviation. The mean and standard deviation have been obtained through 11 Monte
Carlo trials.

     For one episode, the total farm power output of a controller $C$ given by the Algorithm 1 is denoted $P_C = \sum_{t=0}^{H-1} \sum_{i=0}^{N-1} P_t^i$. The metric to benchmark a controller $C$ is the power gain [%] between the total farm power output of $C$ and the total farm power output of the naive controller. The power gain is equal to $100 \cdot \frac{(P_C - P_{\text{naive}})}{P_{\text{naive}}}$.

#### 4.3.1    Perfect predictions

The first set of simulations explores the performance of each controller over increasing variations of wind directions, using perfect predictions. Each state comprises perfect information, i.e., $\epsilon_{K,t} = 0$ and $\epsilon_{V,t} = 0$ for all $t \in \{0, 1, \ldots, 154\}$. The performance of each controller presented in Section 3 is tested for increasing values of $\delta_t^K$.

     Numerical simulations are run on 21 different values of $\delta_t^K$, with $\delta_t^K \in \{0, 1, 2, \ldots, 20\}$. The wind speed is always generated with $\delta_t^V = 1$. Because this work explores the impact of wind direction on wake steering, the magnitude of the wind speed
fluctuations are kept small. The wind direction and wind speed increments are stationary, $\delta_t^K = \delta_{\min}^K = \delta_{\max}^K$ and $\delta_t^V = \delta_{\min}^V = \delta_{\max}^V$ for all $t \in \{0, 1, \ldots, 154\}$.

     The objective here is to study the impact of the wind direction variations on yaw control. The greater the $\delta_t^K$ value, the stronger the variations. Because the nacelles have a limited rotational speed, the study of the wind direction fluctuations is crucial for yaw control. To better illustrate the wind direction dynamics, the time series $\Delta K$ defined as

$$\Delta K = \{|f_{\text{yaw}}(K_{t+1}, K_t)|\}_{t \in \{0, 1, \ldots, 153\}}, \tag{33}$$





is used. Each value of $\Delta K$ lies between $[0, 180]$ [degrees]. To study the magnitude of the variations, the absolute values are taken. Some illustrations of the $\Delta K$ time series for different values of $\delta_t^K$ are given in Figure 8.

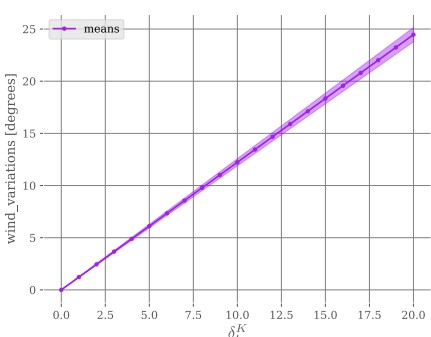 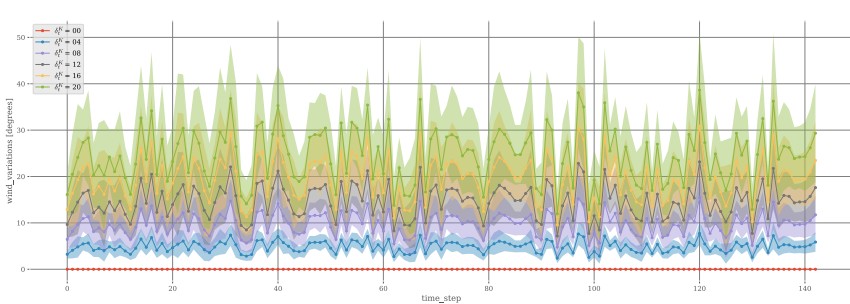

(a) For each $\delta_t^K$, mean values with their standard deviation of the time series $\Delta K$.

(b) Example of time series $\Delta K$ for different values of $\delta_t^K \in \{0, 4, 8, 12, 16, 20\}$. Again, as the $\delta_t^K$ parameter increases, the magnitude of the variations of the wind direction increases.

**Figure 8.** Illustration of the influence of $\delta_t^K$ on the magnitude of wind direction variations $\Delta K$. In Subfigure 8a, the mean value of wind direction variations is given as a function of $\delta_t^K$. In Subfigure 8b, some examples are provided of the time series $\Delta K$ for different values of $\delta_t^K$. For example, for $\delta_t^K = 5$, the mean absolute variations of the wind direction is around 6.11 degrees.

In Figure 9, the power gains of each controller compared to a naive controller are plotted. In Subfigure 9a, the yaw limits are $u_{\min} = \alpha_{\text{cut-in}} = -15$ degrees and $u_{\max} = \alpha_{\text{cut-out}} = 15$ degrees. And in Subfigure 9b, the yaw limits are $u_{\min} = \alpha_{\text{cut-in}} = -30$
degrees and $u_{\max} = \alpha_{\text{cut-out}} = 30$ degrees. These yaw constraints offer enough liberty for a wind turbine to rotate between two consecutive time steps and are small enough to limit the induced fatigue. The detailed results are given in Appendix C, in Tables C1 and C2.

As the variations of the wind direction increase, the performance of each controller diverges from each other. For small variations of the wind direction, both the instantaneous controller and the prediction-based controller give similar results. When
the variations of the wind direction become large, the instantaneous controller struggles to maintain good performance. The heuristic of the prediction-based controller manages to find better yaw control strategies. The gap between the performance of the upper bound with the other controllers shows how strong wind direction variations, in relation with the rotational constraints of each machine, impact yaw control.

As previously said, the time step duration is intentionally imprecise. The reason is that different values of $\Delta t$ will lead to
390 different interpretations. The following statement is true for any values of $\Delta t$, with respect to the hypotheses of the transition regime, described in Subsection 2.6.

– *For wind turbines that can rotate form -15 to 15 degrees every $\Delta t$ minutes, if the wind direction changes by more than 7.34 degrees every $\Delta t$ minutes, it is important to consider future wind data in a steady-state yaw control optimization.*

– *For wind turbines that can rotate form -30 to 30 degrees every $\Delta t$ minutes, if the wind direction changes by more than*
395 *12.23 degrees every $\Delta t$ minutes, it is important to consider future wind data in a steady-state yaw control optimization.*

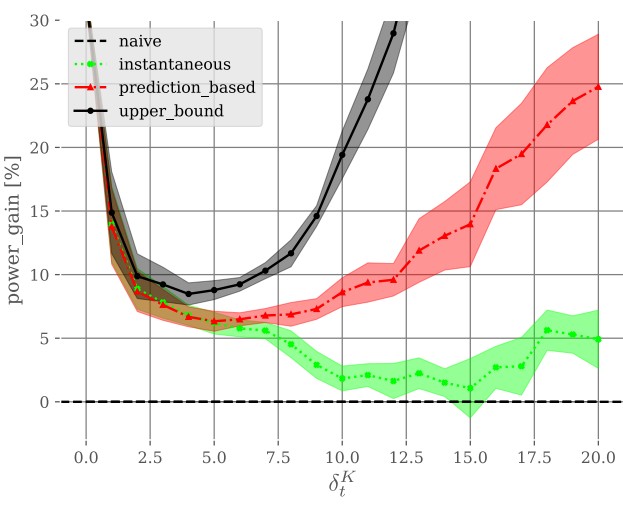
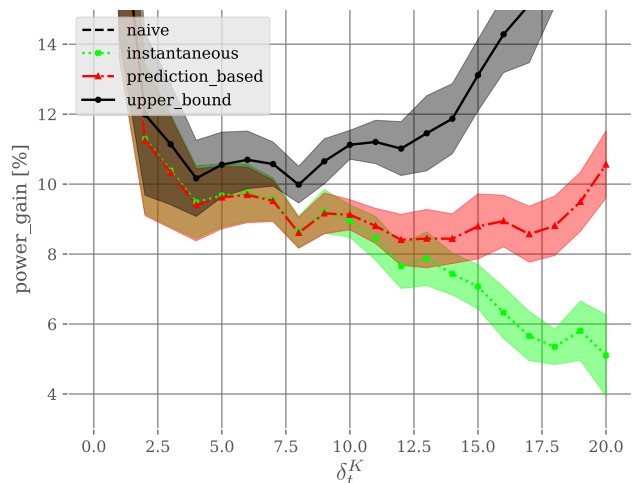

(a) Yaw limits are $u_{\min} = \alpha_{\text{cut-in}} = -15$ degrees and $u_{\max} = \alpha_{\text{cut-out}} = 15$ degrees. At $\delta_t^K = 6$, the prediction-based controller increases the power output of a naive approach by 6.23 %. It corresponds to absolute variations of the wind direction of 7.34 degrees.

(b) Yaw limits are $u_{\min} = \alpha_{\text{cut-in}} = -30$ degrees and $u_{\max} = \alpha_{\text{cut-out}} = 30$ degrees. At $\delta_t^K = 10$, the prediction-based controller increases the power output of a naive approach by 8.93 %. It corresponds to absolute variations of the wind direction of 12.23 degrees.

**Figure 9.** Considering future wind data in a steady-state yaw control optimization becomes mandatory when $\delta_t^k \geqslant 6$ for yaw constraints $[-15, 15]$ degrees and $\delta_t^k \geqslant 10$ for yaw constraints $[-30, 30]$. From these points, the heuristic $\mathcal{H}_t$ provided by the prediction-based controller greatly improves the performance of a classic instantaneous steady-state optimization.

### 4.3.2 Noisy predictions

In the second set of simulations, the robustness to noisy predictions of each controller is tested. The yaw limits are $u_{\min} = \alpha_{\text{cut-in}} = -15$ degrees and $u_{\max} = \alpha_{\text{cut-out}} = 15$ degrees. The $\{K_t\}_{t \in \{0,1,...,154\}}$ time series are computed with $\delta_{\min}^K = 0$ and $\delta_{\max}^K = 20$. The time series $\{V_t\}_{t \in \{0,1,...,154\}}$ are always computed with $\delta_t^V = \delta_{\min}^V = \delta_{\max}^V = 1$. Because $\delta_{\min}^K \neq \delta_{\max}^K$, the incre-

ments are non-stationary for the wind direction. The corresponding $\Delta K$ time series is plotted in Figure 10.

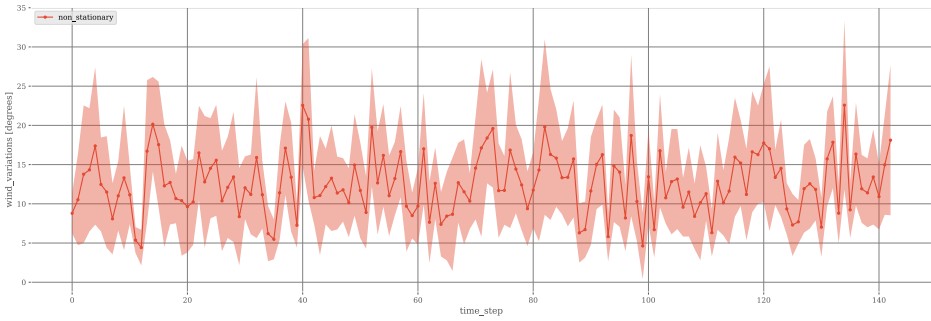

**Figure 10.** Plot of the time series $\Delta K$ for the 11 different seeds. Wind directions are generated with $\delta_{\min}^K = 0$ and $\delta_{\max}^K = 20$. The mean is 12.53 degrees and the standard deviation is 1.12. Here, the increments vary from one time step to another because they are non-stationary.




In Figure 11a, the noise for the wind direction is increasing, i.e., $\epsilon_{K,t} \sim U(-z_K, z_K)$ with $z_K \in \{0, 1, \ldots, 15\}$, for each $t \in \{0, 1, \ldots, 154\}$. The noise for the wind speed is always sampled from the same distribution, i.e., $\epsilon_{V,t} \sim U(-1, 1)$. In Figure 11b, the noise for the wind speed is increasing, i.e., $\epsilon_{V,t} \sim u(-z_V, z_V)$ with $z_V \in \{0, 1, \ldots, 7\}$, for each $t \in \{0, 1, \ldots, 154\}$. The noise for the wind direction is always sampled from the same distribution, i.e., $\epsilon_{K,t} \sim U(-1, 1)$.

Only the noise applied to the wind directions strongly impacts the different policies. The prediction-based controller results in a poorer performance than a naive controller from a noise of 8 degrees. The wind speed noise insignificantly affects the performance of the algorithms. This corroborates the fact that yaw control mainly depends on the wind directions.

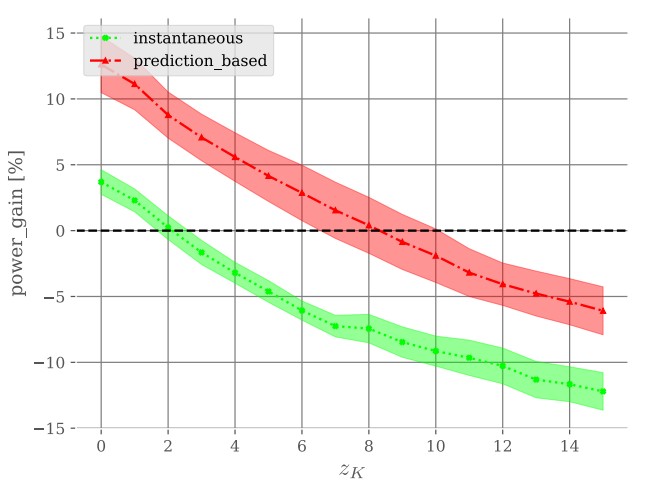
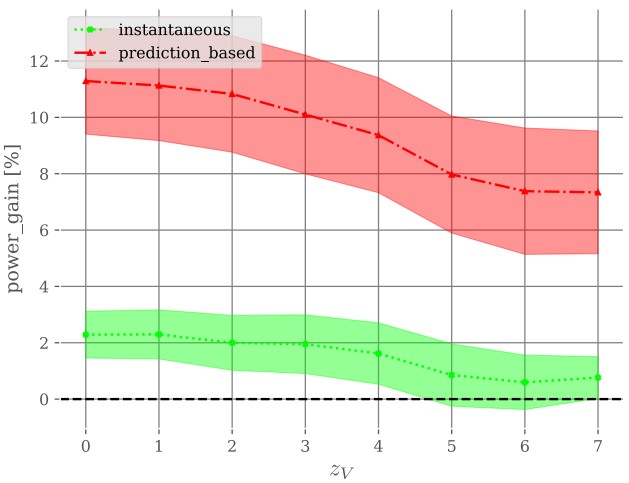

(a) The noises for the wind directions are $\epsilon_{K,t} \sim U(-z_K, z_K)$ with $z_K \in \{0, 1, \ldots, 15\}$, for each $t \in \{0, 1, \ldots, 154\}$. The noise for the wind speeds are always sampled from a Uniform distribution $U(-1, 1)$.

(b) The noises for the wind speeds are $\epsilon_{V,t} \sim U(-z_V, z_V)$ with $z_V \in \{0, 1, \ldots, 7\}$, for each $t \in \{0, 1, \ldots, 154\}$. The noise for the wind directions are always sampled from a Uniform distribution $U(-1, 1)$.

**Figure 11.** Only the noise applied to the wind direction strongly impacts the performance of the algorithms. This corroborates the important dependence between yaw control and the variations of the wind direction. Because the prediction-based controller uses more wind data points, it is more robust than the instantaneous controller.

## 5    Conclusions

As WFFC becomes more important to increase the energy production of wind farms, this work studies wake steering as
a steady-state optimization problem over time. The yaw control problem is formalized as successive multiple steady-state optimization problems. Because each of these states is independent from the others, only the transitions between them really capture the dynamics of the system. The properties governing the transitions are defined as the transition regime and comprise the rotational constraints of the nacelles and the evolution of the wind. Low-fidelity, steady-state simulators are used because they are not time consuming and they are suitable for optimization. Future works could perform the same studies but with





continuous and higher-fidelity simulators such as HAWC2Farm (Liew et al., 2023), better capturing the dynamics of the problem.

Traditionally, yaw control is optimized in a steady-state manner. Yaw settings are computed so that they maximize the instantaneous power output of the farm. To optimize wake steering over a long-term time horizon, an MPC method is usually used. Such an approach increases the complexity of the optimization problem, making it harder to solve. To overcome such

complexity, a reformulation of the steady-state optimization problem is proposed in this work to consider future wind data. The traditional objective function is augmented by a new heuristic estimating the future, expected, theoretical power outputs of the farm, weighted by how far the turbines will be from the wind if they are controlled by a naive approach.

Lastly, this work conducts a sensibility analysis of yaw control and the variations of the wind direction. It demonstrates the importance of optimizing yaw control over future wind data when the variations of the wind directions become large.

For strong wind variations, the new proposed heuristic greatly improves the performance of the controllers without increasing complexity. This work shows for example that if deploying wind turbines that can rotate from -15 to 15 degrees every $\Delta t$ minutes, if the wind direction changes by more than 7.34 degrees every $\Delta t$ minutes, it is important to consider future wind data in a steady-state yaw control optimization. This study is conducted on synthetic wind data so future works should explore the same question of dependence between the wind variations and yaw control over real wind data.

Because the hypotheses regarding the transition regime may be far from reality, the proposed heuristic could be combined with low-pass filters and hysteresis mechanisms for more realistic implementations. Future works should incorporate the fatigue in the optimization process, as WFFC can have a major impact on the lifetime of each turbine. For example, the objective function of the prediction-based controller could be augmented by a heuristic taking into account the magnitude of the yaw actuations. However the results provided by this work also suggest that with wake steering strategies more robust to wind

direction variations, it would be possible to reach the same level of performance with fewer yaw actuations.

## Appendix A:  Gauss-Seidel method

The GS method iterates over each turbine in the direction of the wind, one by one, from upstream turbines to downstream ones. The turbines' default coordinates $\{cx^i, cy^i\}_{i \in \{0,1,...,N-1\}}$ [m] are rotated such that the wind is coming from the west. The initial yaw settings are computed with a naive controller. By doing so, the initial solution is already a good enough solution

that keeps turbines as aligned with the wind as possible. At each iteration, it solves the optimization problem by varying the yaw setting of the current turbine, considering all the others fixed. To solve each optimization problem on one variable, a grid-search approach over a discretized solution space $S$ is used. Once solved, the setting for turbine $k$ is fixed and optimization is conducted again on turbine $k+1$. Such an approach gives good results because it exploits the sequential nature of the low-fidelity simulation.





---

**Algorithm A1** GS method

---

**Input:** $s_t$ input state

$cx_t^i, cy_t^i = (cx^i, cy^i) \cdot \begin{pmatrix} \cos(270 - K_t') & -\sin(270 - K_t') \\ \sin(270 - K_t') & \cos(270 - K_t') \end{pmatrix}$      ▷ rotate turbines

$R = \{i_0, i_1, \ldots, i_{N-1} \mid cx_t^{i_0} \leqslant cx_t^{i_1} \leqslant \ldots \leqslant cx_t^{i_{N-1}}\}$      ▷ order turbines

$(u_{t,0}^{i_0}, u_{t,0}^{i_1}, \ldots, u_{t,0}^{i_{N-1}}) = \pi_{\text{naive}}(s_t)$      ▷ initialize yaw settings

  **for** $i \in R$ **do**

    $S = \{u_{\min} + l \cdot \frac{u_{\max} - u_{\min}}{n_y - 1}\}, \ \forall l \in \{0, 1, \ldots, n_y - 1\}$      ▷ discretized solution space

    $U = (u_{t,1}^{i_0}, u_{t,1}^{i_1}, \ldots, u^i, \ldots, u_{t,0}^{i_{N-2}}, u_{t,0}^{i_{N-1}})$      ▷ fix all other settings

    $u_{t,1}^i \in \underset{u^i \in S}{\arg\max} \ \ f_{\text{obj}}(s_t, U)$      ▷ grid-search optimization

  **end for**

**Output:** $(u_{t,1}^{i_0}, u_{t,1}^{i_1}, \ldots, u_{t,1}^{i_{N-1}})$ yaw settings

---

## Appendix B: Numerical instabilities

First, some modifications have been made to FLORIS in order to shut down those turbines too much misaligned with the wind during a simulation. At a time step $t$, for a given turbine $i$, all the possible yaw settings can give a similar power output. In such cases, the best yaw setting is the one staying the closest to the wind direction, in order to prevent future misalignment. To incorporate such behavior in the optimization process and to make the controllers robust to numerical instabilities, the following steps are taken.

- Round out the power output computed by the simulator.

- Take the yaw setting $u_t^i$ maximizing the objective function.

- Find all the yaw settings giving a performance close to the maximum.

- Among these selected settings, keep the one closest to the setting corresponding to the naive controller.

To compute the solutions of the upper-bound controller described in Subsection 3.3, a trick is used. Relaxing the rotational constraints of the turbines, i.e., making $u_{\min}$ and $u_{\max}$ equal to -180 and 180 degrees, respectively, increases the solution space. With the same precision parameter $n_y$, it reduces the precision of the grid-search method. To keep the solution space between $u_{\min}$ and $u_{\max}$, and therefore, to not alter the precision of the grid-search method, the dynamics of the system described in Algorithm 1 are slightly modified. At the beginning of each time step, every turbine is realigned with the wind direction. Such a trick does not alter the solutions given by the upper-bound controller because good yaw settings are the one keeping turbines close to the wind direction.



## Appendix C: Detailed results

| $\delta_t^K$ | 00 | 01 | 02 | 03 | 04 | 05 | 06 | 07 | 08 | 09 | 10 | 11 | 12 | 13 | 14 | 15 | 16 | 17 | 18 | 19 | 20 |
|---|---|---|---|---|---|---|---|---|---|---|---|---|---|---|---|---|---|---|---|---|---|
| naive | 1.14 | 1.89 | 2.04 | 2.05 | 2.06 | 2.05 | 2.05 | 2.04 | 2.02 | 1.95 | 1.87 | 1.81 | 1.75 | 1.64 | 1.58 | 1.49 | 1.37 | 1.29 | 1.19 | 1.10 | 1.03 |
| instantaneous | 1.49 | 2.12 | 2.20 | 2.20 | 2.21 | 2.18 | 2.17 | 2.14 | 2.10 | 2.00 | 1.89 | 1.85 | 1.78 | 1.67 | 1.60 | 1.49 | 1.40 | 1.32 | 1.25 | 1.16 | 1.08 |
| prediction_based | 1.49 | 2.12 | 2.20 | 2.20 | 2.20 | 2.18 | 2.18 | 2.17 | 2.15 | 2.09 | 2.03 | 1.98 | 1.92 | 1.82 | 1.77 | 1.68 | 1.61 | 1.52 | 1.43 | 1.35 | 1.27 |
| upper_bound | 1.50 | 2.13 | 2.22 | 2.23 | 2.24 | 2.23 | 2.24 | 2.24 | 2.24 | 2.23 | 2.22 | 2.23 | 2.24 | 2.23 | 2.25 | 2.23 | 2.23 | 2.25 | 2.24 | 2.25 | 2.23 |

**Table C1.** Detailed results of the simulations conducted on perfect predictions in Subsubsection 4.3.1. Yaw limits are $u_{\min} = \alpha_{\text{cut-in}} = -15$ degrees and $u_{\max} = \alpha_{\text{cut-out}} = 15$ degrees. For each $\delta_t^K$ are given the total power output of the farm in $10^4$ MW for each controller.

| $\delta_t^K$ | 00 | 01 | 02 | 03 | 04 | 05 | 06 | 07 | 08 | 09 | 10 | 11 | 12 | 13 | 14 | 15 | 16 | 17 | 18 | 19 | 20 |
|---|---|---|---|---|---|---|---|---|---|---|---|---|---|---|---|---|---|---|---|---|---|
| naive | 1.14 | 1.89 | 2.04 | 2.05 | 2.06 | 2.05 | 2.06 | 2.06 | 2.07 | 2.05 | 2.04 | 2.04 | 2.06 | 2.04 | 2.05 | 2.01 | 1.99 | 2.00 | 1.97 | 1.94 | 1.87 |
| instantaneous | 1.65 | 2.18 | 2.25 | 2.25 | 2.26 | 2.25 | 2.25 | 2.25 | 2.25 | 2.24 | 2.22 | 2.21 | 2.21 | 2.19 | 2.20 | 2.15 | 2.11 | 2.10 | 2.07 | 2.04 | 1.96 |
| prediction_based | 1.65 | 2.18 | 2.25 | 2.25 | 2.26 | 2.25 | 2.25 | 2.25 | 2.25 | 2.24 | 2.22 | 2.22 | 2.22 | 2.20 | 2.22 | 2.18 | 2.17 | 2.16 | 2.13 | 2.12 | 2.07 |
| upper_bound | 1.65 | 2.19 | 2.26 | 2.27 | 2.27 | 2.26 | 2.27 | 2.27 | 2.28 | 2.27 | 2.26 | 2.26 | 2.27 | 2.27 | 2.28 | 2.27 | 2.27 | 2.28 | 2.28 | 2.28 | 2.27 |

**Table C2.** Detailed results of the simulations conducted on perfect predictions in Subsubsection 4.3.1. Yaw limits are $u_{\min} = \alpha_{\text{cut-in}} = -30$ degrees and $u_{\max} = \alpha_{\text{cut-out}} = 30$ degrees. For each $\delta_t^K$ are given the total power output of the farm in $10^4$ MW for each controller.

*Author contributions.* EK contributed to the original idea of the heuristic, implemented the codes, conducted all the simulations and wrote most of the paper. PB, FC, PC and DE contributed to the writing and review of the paper.

*Competing interests.* The authors declare that they have no conflict of interest.

*Acknowledgements.* The authors are grateful for the support of the R&D Wind program of TotalEnergies OneTech, especially Cédric ENEAU for his support.

## Acronyms

**FLORIS** FLOw Redirection and Induction in Steady State. 5, 16, 23

**GS** Gauss-Seidel. 10, 17, 22, 23

**IEA** International Energy Agency. 6, 16, 17





**LUT**  lookup table. 2, 3

**ML**  machine learning. 3

**MPC**  model predictive control. 1, 3, 10, 11, 22

**MW**  megawatts. 6, 16, 17, 24

**RL**  reinforcement learning. 3

**WFFC**  wind farm flow control. 2–4, 8, 16, 21, 22





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
