# Peer review of "On the importance of wind predictions in wake steering optimization"

_Wind Energy Science, 2023_

## Author Comment (AC1)

**On the importance of wind predictions in wake steering optimization - response to the reviewers**

Elie Kadoche[1,2], Pascal Bianchi[2], Florence Carton[1],
Philippe Ciblat[2], and Damien Ernst[2,3]

[1]TotalEnergies OneTech, 2 Place Jean Millier, 92400 Courbevoie, France
[2]Télécom Paris, 19 Place Marguerite Perey, 91120 Palaiseau, France
[3]Montefiore Institute, University of Liège, Belgium

We are grateful to the reviewers for dedicating their time to reading the document and providing their insightful remarks. It helped us take a step back and enhance the paper. In the revised paper, the red and green text correspond to the differences between the original (first) and the revised (last) version of the paper. In this document, the colored text in blue and in italic refers to comments of the reviewers. We address each comment individually below.

*Reviewer 1 - The prediction based controller is according to the authors a too difficult optimization problem and simplifications are proposed (top page 12). What is the effect of these simplifications on the original problem statement? Is it possible to quantify these? It feels now like, the original problem is simplified and we will solve the simplified problem. However, we have no idea how far the simplified problem is from the original problem.*

*Reviewer 2 - The most critical suggestion is to motivate and/or justify the simplifications proposed in section 3.2.2. Is it reasonable to ignore locally varying wind speeds and can you quantify the impact? Does assuming a naive controller for the forward-in-time yaw angles have an impact on the current time optimization result (if not, explain)? What is the impact of a cosine vs linear yaw loss model?*

We do agree that the simplifications of the original sequential decision problem made in Section 3.2.2 lack some justifications and quantifications. Because of these simplifications, the solutions are suboptimal. We especially thank the reviewers for their comments regarding this part. As it is one of the key contribution of the paper, we do agree that these justifications need to be addressed in more details.

Unfortunately, these simplifications come from the fact that finding good results for the original decision problem is extremely difficult. We tried different optimization approaches (a massively parallelized genetic algorithm and a double

Gauss-Seidel method) to solve the original decision problem, but without success. We believe that solving such optimization problem is out of reach of our computers.

To quantify the simplifications we propose in Section 3.2.2, we would need to have already obtained good results for the original problem, which we do not have. It is therefore complex to strictly quantify the simplifications, since they are necessary to find a solution in an acceptable computational time. Our proposed simplifications are motivated by some intuitions and by the numerical experiments, that empirically demonstrate their effectiveness. In the paper we identify such study in future work (added to the conclusion) and we give more details regarding the motivations behind each simplification.

**Local wind speeds**  By replacing local wind speeds by the global wind speed, we lose specific spatial information but we keep an overall idea of the potential energy the farm can produce in the future, based on the prediction. And we assume that it is enough to guide future optimization towards good solutions.

**Future yaw angles**  By replacing the wake steering optimization performed in the future by a naive wind tracking solution, we can keep an overall idea of the future orientations of each turbine. A wind tracking solution can be computed easily and will give good enough solutions in average. Because we know that good solutions will be close to the wind.

**Cosinus**  The heuristic is $\mathcal{H}_t(s_t, u_t) = \sum_{k=1}^{L} \gamma^{k-1} \cdot \Phi(V'_{t+k}) \cdot h(\hat{\alpha}^i_{t+k+1}{}')$, with $\hat{\alpha}^i_{t+k+1}{}'$ the future expected yaw angle in degrees (between -180 and 180 degrees). The function $h$ returns a penalization term for the yaw: it is 1 if the yaws are equal to 0 degrees (perfect alignment) and it is 0 if the yaws are equal to -180 or 180 degrees (most extreme misalignment).

- By keeping the original decision problem (no simplification), we would have $h(\hat{\alpha}^i_{t+k+1}{}') = \frac{1}{N} \cdot \sum_{i=0}^{N-1} \cos^p(\frac{|\hat{\alpha}^i_{t+k+1}{}'|}{2} \cdot \frac{\pi}{180})$.
- In the paper, we decide to change the function such that $h(\hat{\alpha}^i_{t+k+1}{}') = \left(1 - \frac{1}{N} \cdot \frac{1}{180} \cdot \sum_{i=0}^{N-1} |\hat{\alpha}^i_{t+k+1}{}'|\right)$.

We run some experiments with both versions. The linear penalization (ours) gives slightly better results (2 % more power in average) than the cosine penalization (original). Our intuition is that the linear approximation gives equal penalization for every yaw values (see plot for one turbine below).

[Figure]

Figure 1: Penalization functions for one turbine, with $p = 1.88$.

**Indicator function**  The indicator function is removed so that even with important yaws, we have some idea of how far away from the wind each machine is.

**Discount factor**  And a discount factor is added to give move importance to immediate time steps. It is common practice for model predictive control based optimization.

**1  Anonymous Referee #1**

*This paper describes the application of different controllers that maximize the power output of a wind farm by optimizing the turbines yaw angles under time-varying wind directions. Different control strategies are tested and the results are compared. I have the following remarks/comments:*

*- The authors realize that a wind farm is a dynamical system (mainly due to wake delays). However, the authors try to push a steady-state model in a dynamical control framework. Isaac Newton went through great lengths to introduce differential equations, why not use these in this application? Why not model the windfarm using differential equations? In fact, in algorithm 1, at each time step, the state $s_t$ is evaluated and new control signals are computed accordingly. However, the wind farm will never reach the computed state $s_t$ since this is a steady-state of the farm and control signals are set at each time step and the wind velocity is changed at each time step. So what is actually optimized here?*

We especially thank the reviewer for this comment as it was not clearly explained in the first version of the paper (we added more details in the revised version). There is a distinction between the simulation (steady-state) and the overall

system dynamics (not steady-state). The overall system dynamics is not steady-state because of the rotational constraints of the machines and the evolution of the wind. But at each iteration, to compute the power output of the farm, a steady-state simulation is performed.

In wind farm optimization, the use of steady-state models (known as low fidelity simulation) like FLORIS is favored over differential equations due to the complexity and computational load associated with solving dynamic equations for every turbine in the farm. Differential equations would require accounting for fluid dynamics, turbulence, and wake interactions in real-time, making it impractical for optimization purposes (but these models exist and are used for other purposes, it is high fidelity simulation). Steady-state models like FLORIS simplify these dynamics by approximating the wind flow within the farm under steady conditions, allowing for efficient optimization algorithms to be applied for control strategies.

*- In the abstract, at the end, the authors write "it does not increase complexity". This seems to be a relative notion. What does complexity mean here and does it become more complex for everyone?*

We do agree that this a relative notion. By "complexity", we mean computational time, which is also relative to the computational power of each person. But overall, as there are more turbines, the computational complexity explodes for every machines. The heuristic we propose in the paper is quasi-instantaneous, for any number of turbines, making the overall reformulated optimization problem, simpler to solve. The "complexity" also refers to the number of decision variables, that the proposed heuristic also manages to reduce.

*- It seems that the wind speed in front of turbine $i$ is defined as $v_t^i$ which is later defined as $K_t$. I would recommend to take out unnecessary variables to make the document easier to follow.*

The global wind direction is $K_t$ and the global wind speed is $V_t$. The wind speed if front of a turbine is defined as $\nu_t^i$. But the wind direction in front of a turbine ($\kappa_t^i$ for example) is never defined. Because the wind direction in front of the turbines is always equal to the global wind direction, i.e., $\kappa_t^i = K_t \forall i \in \{0, 1, \ldots, N-1\}$. It is clarified in the revised version.

*- In Figure 2, is it possible to also indicate $u_t^i$?*

Absolutely, we added it to the figure.

*- In (4), what does the one at the end of the equation mean?*

The indicator function, denoted as $\mathbb{1}_A(x)$, is a mathematical function that takes the value 1 if its argument belongs to a specified set $A$, and 0 otherwise. Formally,

it is defined as $\mathbb{1}_A(x) = \begin{cases} 1 & \text{if } x \in A \\ 0 & \text{if } x \notin A \end{cases}$. It serves as a simple way to represent whether a certain condition or event occurs. In the paper, we use this notation to check if the yaw of the machine belongs to the safety bounds.

*- In (5), the notation is not clear. There is no function defined, but only ... Please detail this.*

We added more details regarding the state. It contains the current wind $(K_t, V_t)$ and the future predicted wind on $L$ time steps: $(K_{t+1}, V_{t+1}), (K_{t+2}, V_{t+2}), \ldots, (K_{t+L-1}, V_{t+L-1}), (K_{t+L}, V_{t+L})$.

*- The term MPC is used in the paper. However, the controller is clearly not an MPC. I would suggest taking out the term to avoid confusion.*

We thank the reviewer for this comment, the first version was unclear regarding this subject. We clarified this by saying: "the corresponding original sequential decision problem over a future time window can be stated under a form usually exploited by the MPC community". And by replacing the occurrences of "MPC" by "original decision problem" or "prediction-based, sequential decision problem".

*- What is the relation between f_control, f_yaw and $\pi(s_t)$. Is it possible to simplify notation? It seems overcomplicated, but maybe it is really necessary like this?*

We put a lot of effort into the notation so that it is clear and precise enough. The relation between the different functions are described in Algorithm 1. We tried to simplify a little more the paper regarding the notation, but a certain level of complexity is necessary.

*- In (12), what does clip() mean?*

The "clip()" function is a mathematical operation commonly used in programming libraries. It takes three arguments: a value $x$, a lower bound $a$, and an upper bound $b$, and returns a new value that is clipped to fall within the range $[a, b]$. Mathematically, it can be expressed as $\text{clip}(x, a, b) = \min(\max(x, a), b)$. The "clip()" function ensures that the value $x$ remains within the specified range, preventing it from exceeding either the lower or upper bound.

*- Around 235 the authors write "At each iteration, it solves the optimization problem for the current turbine, considering the yaw angles of all others fixed. To do so, it uses a grid-search method..." What is now done in the end? A grid search or is an optimization problem solved?*

We do agree that this part was not clear. We improved the paper to better explain the overall process.

*- The prediction based controller is according to the authors a too difficult optimization problem and simplifications are proposed (top page 12). What is the effect of these simplifications on the original problem statement? Is it possible to quantify these? It feels now like, the original problem is simplified and we will solve the simplified problem. However, we have no idea how far the simplified problem is from the original problem.*

Unfortunately, we cannot quantify these simplifications, because we were not able to get good solutions for the original decision problem. We addressed this answer in the first page of the document.

*- In Figure 4, please indicate better the meaning of all symbols/lines and the wind direction.*

The Figure 4 was indeed not very clear. We completely change it to improve readability and to better illustrate our proposed heuristic.

*- In line 337 I read that some option are enabled. What does this mean?*

It corresponds to specific wake modelling options related to the FLORIS software, that provide additional features to the low fidelity simulation. We specified them in the revised paper for reproducibility purpose.

*- Overall, please provide tables with settings that are used throughout the simulations. These are now everywhere placed in the text which makes it for me impossible to follow.*

Absolutely, we added tables with all the parameters used across the simulations.

*- In line 350, L=11 is defined. What does this mean in the context of steady-state models?*

The variable $L$ corresponds to length of the future time window a controller has access to. We added the following sentence in the paper: "for example, if $\Delta t$ corresponds to 5 minutes, then the horizon $L = 10$ means that the prediction-based controller has access to a prediction of the wind of 50 minutes."

*- In figure 9, how is the upper bound computed? how are the shaded areas computed?*

The upper bound is computed with the rotational constraints relaxed. At each time step, the solution should be quasi-optimal. The shaded ares correspond to the standard deviation of the numerical experiments results. We added more explanations regarding this in the revised paper.

*- In line 411, the authors write "capture the dynamics of the system". I don't think that this is correct since a steady-state model is used.*

The overall system dynamics is not steady state. Time steps are interconnected by the evolution of the wind and the rotational constraints of the wind turbines. So it captures the dynamics of: the global wind evolution between time steps and the yaw rotational constraints between time steps.

Only the simulation function $f^i_{\text{simulation}}$ is steady-state: at each time step, the computation of the wake effects and the evolution of the wind across the machines is based only on the current wind data only. We detailed this in the revised version.

*- Figure 3 seems to be not necessary? This is a well known figure, but what does it contribute to the story?*

Absolutely, it is not necessary as it only makes things heavier. We removed it in the revised version.

*The major question that I have is regarding the use of a steady-state model in a dynamical control framework. It raises many questions and the meaning of the results is not clear to me. In other words, how can anybody judge the scientific relevance of the work? I would also suggest the authors to also rewrite the paper so that it becomes more readable/understandable. Define all variables clearly, figures.*

Steady-state models are necessary for optimization. Higher fidelity models, taking into account the dynamics of the wind and the variations of the wake effects between time steps, exist. But these models are too computationally expensive to be used for optimization.

Then, we use a low-fidelity simulator, computing the wake effects in a steady-state manner. This low-fidelity simulator is then incorporated to our full system dynamics, which is not steady-state. Only the simulation, i.e., the function to compute the power output is steady-state. The dynamics of the wind and the yaw angles of each turbine are not steady-state.

We believe that with such framework, our work is scientific relevant and that our work demonstrates an interesting reformulation of a well-known optimization process. However, the amount of uncertainty regarding our results is important because of the use of a steady-state simulation to compute the power outputs. Empirical simulations give an order of magnitude of the added value of our reformulation but need to be performed on higher fidelity models to better asses the added performance.

*I hope that the above remarks can help and I am looking forward to a revised version.*

**2    Anonymous Referee #2**

*General Comments. Overall, the paper is nicely formulated and presented. The problem statement is well defined, and the proposed solution is motivated, described, and validated well. I appreciate the pace of developing the wind farm flow control domain, the problem at hand (costly MPC-based optimization) and the proposed improvement (heuristic-based optimization). In general, I am convinced by the scientific method used, and I am interested to continue to understand the proposed control algorithm.*

*The paper would generally benefit from an effort to improve the flow and readability. Some statements are made without reference to their background or context. Also, there is possibly an excess of mathematical notation in the narrative content.*

We tried to simplify a little more the paper regarding the notation, but a certain level of complexity is necessary. We simplified some terms, we improved the figures, we gave more explanations regarding the steady-state characteristic of the simulation, and we gave more details regarding the results.

*The most critical suggestion is to motivate and/or justify the simplifications proposed in section 3.2.2. Is it reasonable to ignore locally varying wind speeds and can you quantify the impact? Does assuming a naive controller for the forward-in-time yaw angles have an impact on the current time optimization result (if not, explain)? What is the impact of a cosine vs linear yaw loss model?*

We do agree that we should better motivate the justifications we propose in Section 3.2.2. We added more details regarding our motivations in the revised paper. Unfortunately, we cannot quantify these simplifications, because we were not able to get good solutions for the original decision problem. We addressed this answer in the first page of the document.

*With that clarification and a few notes below, I think this paper will be a strong study of an improved controls optimization algorithm.*

*Specific comments. See above for the questions on the motivation and justification for the simplifications to the MPC-based method.*

*Section 3.2.2 presenting a common MPC method - is the formulation a common or typical formulation? Suggest to reference.*

We thank the reviewer for this comment, the first version was unclear regarding this subject. We clarified this by saying: "the corresponding original sequential decision problem over a future time window can be stated under a form usually exploited by the MPC community". And by replacing the occurrences of "MPC" by "original decision problem" or "prediction-based, sequential decision problem".

*Figure 4 is difficult to understand. What are the symbols and how do I know which (a or b) is better?*

The Figure 4 was indeed not very clear. We completely change it to improve readability and to better illustrate our proposed heuristic.

*Line 392: These are important statements, but they seem to come suddenly and the values aren't traceable. It would help to derive these results or relate to Figure 9. Also, consider mentioning section 4.1 or making the "tau x delta-k" statement into an equation that you can reference back to here.*

Absolutely, we improved the readability of the results accordingly.

*Technical corrections. Apologies if it is intentional, but it's unclear if you intended "sensibility analysis" or "sensitivity analysis".*

We thank the reviewer for this remark. We had some doubts about the right term to use. But indeed, the right term is "sensitivity analysis". We updated the paper accordingly.

*Line 42: Suggest to replace "wind direction" with "wind direction variation" to note that it's the change in wind direction that you're studying.*

Absolutely.

*Line 43: Suggest to motivate the use of steady-state models. You did this in the conclusions, but it would be helpful in the intro.*

Absolutely, we added more information regarding that subject in the introduction.

[revised manuscript text omitted]

---

## Author Response (AR2)

**On the importance of wind predictions in wake steering optimization - response to the reviewers (bis)**

Elie Kadoche[1,2], Pascal Bianchi[2], Florence Carton[1],
Philippe Ciblat[2], and Damien Ernst[2,3]

[1]TotalEnergies OneTech, 2 Place Jean Millier, 92400 Courbevoie, France
[2]Télécom Paris, 19 Place Marguerite Perey, 91120 Palaiseau, France
[3]Montefiore Institute, University of Liège, Belgium

We thank the reviewers for taking the time to read our first revised paper. In the second revised paper, the red and green text correspond to the differences between the first revision and the second revision (last and current version). In this document, the colored text in blue and in italic refers to comments of the reviewers. We address each comment individually below.

**1 Anonymous Referee #1**

*I think that there is a misunderstanding about what dynamics are. If a signal (like the wind or yaw angles) is changing over time, it does not mean that dynamics are taken into account. Dynamics are taken into account when there is a model that takes as an input the time-varying signals and outputs another signal (like turbine power). This output signal is then changing due to the input signal, and its exact change depends on the model dynamics. Delays and inertia are typical phenomena that you model with a dynamical model and clearly present in the application at hand and not taken into account.*

We thank the reviewer for insisting on that matter. Indeed, the term "dynamics" implies several phenomena that we are not taking into account in the simulations. Instead, we describe a temporal evolution of multiple states, interconnected via the rotational constraints of the machines. We corrected that in the revised paper.

*The authors write: "Steady-state models are necessary for optimization. Higher fidelity models, taking into account the dynamics of the wind and the variations of the wake effects between time steps, exist. But these models are too computationally expensive to be used for optimization."*

*The statement that steady state model are necessary is questionable. It is a (over?) simplification of the reality and consequently, certain optimization problems can be solved. However, dynamical models can also be used in optimization problems, like high fidelity models with CPU time as a challenge. These things are known/researched, but where is the middle ground? A computational efficient enough dynamical model that can be used in an optimization. This seems to me the important and open question. Using a steady state model in some dynamical control framework remains questionable for me and the authors did not really convince me with their answers.*

We agree that the application of steady-state models in wake steering optimization has important limitations. Finding a computational efficient enough dynamical wake model that can be used in optimization is the focus of active research in the community. But this subject goes outside the question raised in this paper. We focus on the optimization process itself, adhering to community standards by using widely accepted, open-source, low-fidelity simulators. We propose an improvement of a well known optimization problem, using state of the art steady state simulators.

We agree that the sentence "steady-state models are necessary for optimization" is badly formulated, we removed it in the revised paper. We better explained the necessity for steady-states models. An important challenge of wake steering optimization is to develop closed-loop controllers, able to perform continuous optimization, based on feedbacks of the environment (wind data, turbine orientations, power outputs, etc.). Conducting model-based, closed-loop optimization as the farm is operating, requires a simulator of the environment. In that context, high-fidelity simulators are not usable for large (offshore) wind farms because they require too much computational time. Instead, we use lower-fidelity simulators, which offer computation times fast enough to be used in a closed-loop controller.

*At last, the authors state that the original optimization problem is not solvable. Why not perform a grid search? You can let the CPU do the work. If the search space is too large (which I doubt when taking a relatively small farm and relatively small prediction horizon), you can limit it around the optimal solution found by the simplified optimization problem.*

We thank the reviewer for this comment. Indeed, the sentence "the optimization problem is not solvable" is badly formulated. To our knowledge, the optimization problem seems to be difficult to solve in a polynomial time. We corrected that in the revised paper.

**Original problem**   In the original optimization problem, the number of turbines is 34 (this is a relatively small wind farm), the solution space for the yaw of one turbine in discretized in 120 values and the horizon is 10 data points. In this configuration, the total number of scenarios is equal to $120^{34^{11}} = 120^{374}$.

**Simplified problem**  In a simplified but interesting enough problem, the number of turbines should still be 34 (this is already a small wind farm), the solution space for the yaw of one turbine could be discretized in 30 values and the horizon could be composed of 3 data points. In this configuration, the total number of scenarios is equal to $30^{34^4} = 30^{136}$.

On an Apple M2 Pro MacBook Pro with 32 GB of memory, the computation time for 1 scenario is about 0.24 seconds. Leveraging the vectorization capabilities of the simulator, we can reach 193.74 seconds for $10^4$ scenarios. Given that exploring $10^{11}$ scenarios should take approximately 61 years, a grid-search approach does not seem practical.

**2  Anonymous Referee #2**

*This is a much improved version of the paper. Well done considering and incorporating the previous feedback. I especially appreciate the added clarity in describing the future power heuristic and the relation back to applied problems in the conclusion.*

*My only feedback is that Figure 3 is still not clear. The description of the heuristic in Section 3.2.2 is good, but I'm not able to map this to the figure. Consider the following suggestions:*

- *Change the black arrow that points to a label to a different type of arrow so that it does not look like the other arrows that indicate some distance.*
- *I'm not sure what the "rotation zone" represents - is it the possible cone for the wake centerline based on + / - 15 degrees yaw?*
- *In the description, it says "to get an average idea of how far the turbine will be from the predicted wind direction". It's difficult to understand what is meant by a distance from the wind direction.*
- *Is it possible to represent the turbine by a yawed line rather than a "x"?*

We thank the reviewer for the positive feedback. We updated the Figure 3 according to the proposed suggestions. The distance from the wind direction is actually the yaw of the machine, it is the angular distance between the turbine orientation and the wind direction. This is better explained in the revised paper. The rotation zone represents the cone of the future possible orientations of the turbine, centered around the current turbine orientation. This is detailed in the revised paper.